# Whole Flour of Purple Maize as a Functional Ingredient of Gluten-Free Bread: Effect of In Vitro Digestion on Starch and Bioaccessibility of Bioactive Compounds

**DOI:** 10.3390/foods13020194

**Published:** 2024-01-06

**Authors:** Luisina Monsierra, Pablo Sebastián Mansilla, Gabriela Teresa Pérez

**Affiliations:** 1Facultad de Ciencias Agropecuarias (FCA), Departamento de Agroalimentos, Universidad Nacional de Cordoba (UNC), Ing. Agr. Felix Aldo Marrone 746, Cordoba 5000, Argentina; lmonsierra@agro.unc.edu.ar (L.M.); pmansilla@agro.unc.edu.ar (P.S.M.); 2Instituto de Ciencia y Tecnología de los Alimentos Córdoba (ICYTAC), CONICET-UNC, Avenida Filloy s/n, Cordoba 5000, Argentina; 3Facultad de Ciencias Agropecuarias (FCA), Cátedra de Química Biológica, Departamento de Fundamentación Biológica, Universidad Nacional de Cordoba (UNC), Ing. Agr. Felix Aldo Marrone 746, Cordoba 5000, Argentina

**Keywords:** *Zea mays* L., antioxidant capacity, polyphenols, anthocyanins, dialysability

## Abstract

The growing demand for gluten-free products requires the study of alternatives to produce nutritionally and technologically favorable foods. The aim was to evaluate the content and antioxidant capacity of gluten-free bread enriched with whole flour of purple maize (PM) and how starch and bioaccessibility of antioxidant compounds were modified during in vitro digestion. Gluten-free bread was prepared with the addition of 34%, 50%, and 70% PM, and white maize bread served as control. The content of total polyphenols, anthocyanins, and antioxidant capacity through FRAP and TEAC was measured. Specific volume, crumb texture, and starch digestibility were determined in the breads. Simultaneously, in vitro digestion and dialysis by membrane were performed to evaluate the bioaccessible and potentially bioavailable fraction. Bread with 34% PM had a similar specific volume and crumb texture to the control, but higher content of polyphenols (52.91 mg AG/100 g), anthocyanins (23.13 mg c3-GE/100 g), and antioxidant capacity (3.55 and 5.12 µmol tr/g for FRAP and TEAC, respectively). The PM breads had a higher antioxidant content and capacity and higher slowly digestible and resistant starch than the control. These parameters increased as the PM proportion rose. After digestion, anthocyanins were degraded, polyphenols and antioxidant capacity decreased, but they remained potentially bioavailable, although to a lesser extent. Bread with 34% shows acceptable technological parameters, lower starch digestibility, and contribution of bioactive compounds with antioxidant capacity. This indicates that purple maize flour represents a potential ingredient to produce gluten-free bread with an improved nutritional profile.

## 1. Introduction

Gluten-free products are increasingly demanded as a result of the growing worldwide incidence of gluten-intake-related diseases and the belief that they are associated with a healthier lifestyle [1]. A distinguishing feature of these products is the absence of proteins essential for forming a network that traps fermentation-derived carbon dioxide. Therefore, there is a need to study new alternatives for the industrial production of gluten-free foods that offer nutritional benefits and have technological characteristics similar to those of their gluten-containing counterparts [2].

Purple maize is widely cultivated and consumed in Mexico, the United States, the Andean region, and South America, mainly in Peru, Bolivia, Ecuador, and northern Argentina. It features a characteristic color and high content of phenols in the pericarp, mainly anthocyanins [3]. Numerous studies associate these compounds with potential health benefits that include antioxidant, anti-inflammatory, antimutagenic, anticarcinogenic, and anti-angiogenesis capacity, as well as hypoglycemic activity and prevention of obesity and cancerous diseases [4,5]. These advantages are only possible if the compounds reach their target site (organ or cells), being first bioaccessible and subsequently bioavailable to be used by the human body. Bioaccessibility is defined as the maximum fraction of a substance that can be dissolved and released in the gastrointestinal tract from the original food matrix [6]. Bioavailability is understood as the fraction of a substance that is absorbed by the gastrointestinal tract and reaches the systemic circulation [7]. These processes can be affected by the composition and characteristics of the food matrix, such as solubility, pH, composition, viscosity, and technological treatments, and factors of the person who ingested the food, such as sex, age, and microbiota composition [8].

Starch is the major component of the maize grain. Starch digestibility can be classified as rapidly digestible starch (RDS), slowly digestible starch (SDS), and resistant starch (RS), according to the rate and extent of in vitro digestion. The RDS promotes rapid increases in blood glucose and insulin levels in humans, while the SDS fraction generally provides a slow and prolonged release of glucose into the bloodstream. The reduction of starch hydrolysis rate and the increase in RS can help lower blood sugar levels, thereby preventing and interfering with the physiological functions of chronic diseases [9]. Polyphenols are associated with these benefits because they can inhibit digestive enzymes by binding to their active site. In this way, they compete with glucose for the transporter during absorption by the enterocyte and/or by the formation of networks during starch gelatinization, which modifies its structure and functionality [10].

The elimination of gluten has an impact on food quality attributes, nutritional characteristics, and consumer acceptance. Generally, bakery products for celiac patients have inferior dough rheological properties, often lacking cohesion and elasticity and featuring less desirable texture and quality [11]. In addition, the ingredients generally used to make them tend to be high in carbohydrates and calories and lower in protein than their gluten-containing counterparts. Therefore, a balance must be found between the nutritional and technological quality of gluten-free products [12].

For all this, when considering new alternatives to raw materials for gluten-free food making, purple maize flour could be used due to its functional properties. However, in order to determine the nutritional quality of the final product, it is important to assess the effect of food processing and the influence of digestion on the bioavailability and functionality of bioactive compounds, as well as the starch hydrolysis rate.

Therefore, the objective of this study was to evaluate the effect of adding whole flour of purple maize to the antioxidant content and capacity of gluten-free bread, how this affects the in vitro digestion of starch, and the bioavailability of antioxidant compounds present in the bread.

## 2. Materials and Methods

### 2.1. Flour Samples

Whole purple maize flour (PM) was obtained from the “Moragro’’ cultivar, an open-pollinated variety adapted to the central semiarid region of Córdoba (Argentina), registered before the Instituto Nacional de Semillas (INASE). In this work, seeds from the 2019/20 campaign were used. The assay was conducted at the experimental station of Facultad de Ciencias Agropecuarias, Universidad Nacional de Córdoba (31°28′49.42″ S and 64°00′36.04″ W). The seeds were sown and planted in a field located in the province of Córdoba, the central semiarid region of the country, at an altitude of 425 m.a.s.l. The area has a historical average range of medium, minimum, and maximum temperatures of 15 °C–20 °C, 8 °C–13.7 °C and 21.8 °C–25.1 °C, respectively, and annual precipitation of 300 to 1000 mm [13]. The grains were ground in a cyclone mill (Cylotec CT193, Foss, Suzhou, China) without prior conditioning to obtain whole meal flour (particle size less than 500 μm). Commercial rice (RF) and cassava (CF) flours were used in the bread formulations, and white maize whole meal flour (WM) was used as a control to compare with purple maize bread.

### 2.2. Chemical Composition

The moisture, protein, lipid, and ash content of the raw flours used in baking (purple maize, white maize, cassava, and rice) were determined using the Approved Methods of the American Association of Cereal Chemists International [14]. All determinations were made at least in triplicate and expressed as g per 100 g (%) of sample on dry weight (dw).

### 2.3. Viscosity and Flour Pasting Properties

The changes in viscosity by heating individual flour samples and mixtures thereof were determined using a Rapid Visco Analyzer (RVA-4500, Perten Instruments, Springfield, IL, USA) through the RVA standard method. Mixtures of flours were analyzed to determine the interaction among them during heating. Three grams of sample were transferred to the RVA vessel, and 25 mL of distilled water was added. The suspension was stirred at 160 rpm while heating to 50 °C, and it was kept at this temperature for 1 min. Then, the temperature was increased to 95 °C at a heating rate of 9.4 °C/min, with a stirring rate of 960 rpm. The suspension was kept at 95 °C for 2.5 min, and finally, it was cooled at a rate of 11.8 °C/min to 50 °C. Pasting temperature (PT), peak viscosity (PV), final viscosity (FV), trough viscosity (TV), breakdown (BD), and setback (SB) were obtained from the pasting curves. All determinations were made at least in triplicate, and the results were expressed in centipoises.

### 2.4. Bread Making

The basic formulation of control bread was 33% of RF, 33% of CF, 34% of WM, 2% of salt, 2% of vegetable shortening, 4% of compressed yeast, 78% of water, and 0.25% of CMC (carboxymethylcellulose). WM was replaced with PM at a proportion of 34%. In formulations, this substitution was made at a proportion of 50% PM and 70% PM. In all cases, the ingredients were mixed and kneaded for 2 min with a table mixer (Peabody SMARTCHEF PO-BMP 19R, 2,0 HP of power, Buenos Aires, Argentina). Then, the dough was arranged into separate aluminum cups (70 g) and placed in a fermentation chamber for 1 h at 30 °C and 85% relative humidity. To evaluate the fermentation effect, pre-fermented (PRE-F) and post-fermented (POST-F) dough fractions were separated, freeze-dried, and used to perform the same determinations that were made on flour and bread. Finally, the rest of the dough samples were baked in a rotary oven (Beta 107 IPA, Pauna, Buenos Aires, Argentina) for 30 min at 180° to obtain the bread. The entire procedure was carried out in duplicate.

The bread-specific volume (SV) was determined by the displacement of rapeseeds and was calculated as the volume-to-weight ratio of the sample. Five determinations were made per batch, and the results were expressed in cm^3^/g.

A crumb texture (CT) analysis of the bread was performed at 0, 24, and 72 h, according to Approved Methods of the American Association of Cereal Chemists International [14], using a texture analyzer (Instron Model 3342 Universal Testing Machine, Canton, OH, USA). The results were expressed in Newton (N).

### 2.5. Determination of Anthocyanins, Polyphenols, Ferulic Acid, and Antioxidant Capacity

The extraction of antioxidant compounds was carried out by blending 150 mg of sample (flour, pre- and post-fermented dough, and bread) with 1.5 mL (1:10) of ethanol (96%)/HCl (1N) (85:15, *v/v*) and stirred for 30 min at room temperature. Subsequently, the samples were centrifuged (Thermo Fisher Scientific, Sorval ST40R, Madrid, Spain) at 8000× *g* for 10 min, and the supernatant was recovered. The extraction was repeated 3 times, and the supernatants were pooled.

Total anthocyanin content (TAC) was determined by a differential pH method, according to Lee et al. [15], using the calculation formula:$$Anthocyanin\;pigment\;\left(cyanidin-3-glucoside\;equivalents\right)=\frac{A\times MW\times DF\times 10^{3}}{\varepsilon \times 1}$$
where *A* = ${(A}_{520\;nm}-A_{700nm})pH\;1.0-{(A}_{520\;nm}-A_{700nm})pH\;4.5;\;$*DF* = dilution factor; using a molar extinction coefficient (ε) of 26,900 L/mol/cm and a molecular weight (*MW*) of 449.2 g/mol for cyanidin-3-glucoside. The absorbance at 520 and 700 nm was measured with a spectrophotometer (UV-vis JascoV-730, Jasco Corporation, Tokio, Japan). The results were expressed in mg of cyanidin-3-glucoside equivalent per 100 g of sample (mg c3-GE/100 g). Total polyphenol content (TPC) was measured using the Folin-Ciocalteu method, as adapted by Lopez-Martinez et al. [16]. The results were expressed in mg of gallic acid per each 100 g of sample (mg AG/100 g), using a standard curve from 0 to 0.5 mg AG/mL. Ferulic acid content (FA) was measured according to Podio et al. [17] with a modification proposed by Navarro et al. [18]. Briefly, after alkaline hydrolysis and acidification treatment, the samples were centrifuged at 16,000× *g* for 20 min at 4 °C. Three milliliters of ethyl acetate were added to the supernatant and mixed at 6000 g for 15 min. The ethyl acetate phase was recovered from the resulting multilayer system formed. This procedure was repeated to complete three ethyl acetate washes. Then, sodium sulfate was added to the organic phase, stirred, and kept for 1 h in the dark, and the absorbance at 320 nm was measured with a spectrophotometer (UV-vis JascoV-730, Jasco Corporation, Japan). The results were calculated with linear regression, using acid ferulic as standard and expressed in μg of acid ferulic per g of sample (μg FA/g).

The antioxidant activity was quantified by free radical scavenging capacity through a TEAC (Trolox Equivalent Antioxidant Capacity) assay, according to Re et al. [19]. The reducing power was determined with a FRAP (Ferric Reducing Antioxidant Power) assay, according to Benzie and Strain [20]. The results were quantified from a trolox calibration curve and expressed in µmol of trolox per g of sample (µmol tr/g). All determinations were analyzed at least in quadruplicate.

### 2.6. In Vitro Static Digestion

An in vitro digestion model was developed according to the procedure described by INFOGEST’s scientists [6]—as adapted by Bustos et al. [7]—where oral, gastric, and intestinal phases were analyzed. In short, during the oral digestion phase (2 min), chewing was simulated with the addition of salivary fluid (50/50 *w/v* food/simulated salivary fluid) and α-amylase (750 U/mL, A3176 Sigma Aldrich, Merck KGaA, Darmstadt, Germany). In the stomach digestion phase (at 2 h), 1M of hydrochloric acid was added to the gastric fluid until pH 3 was reached, followed by pepsin (25,000 U/mL, P7000 Sigma). Finally, in the intestinal digestion phase (2 h later), 1 M of sodium bicarbonate was added until the sample reached pH 6–7. Then, pancreatin (16 mg/mL, P7545 Sigma) and bile salts (160 mM, Sigma B8756) were incorporated. The samples were put into a shaking water bath at 150 rpm and 37 °C to simulate the digestion process contractions. Finally, a dialysis membrane (MWcutoff = 10 kDa) with 1M of sodium bicarbonate was added and incubated for 3 h.

#### 2.6.1. Starch Digestion

Total starch content was determined using a resistant starch kit (Megazyme International Ireland, Bray, Ireland) as described by Mansilla et al. [21]. Total starch content was considered as resistant starch plus non-resistant starch. The values obtained were used as a reference to analyze starch digestion.

During digestion, aliquots of 1 mL were taken at time 0 of oral, gastric, and intestinal phases and then at 10, 20, 30, 60, and 120 min to monitor starch hydrolysis. The reducing sugar content was determined by the 3,5-dinitrosalicylic acid (DNS) method [7]. The RDS value was obtained from the glucose released after 20 min. SDS was measured from the glucose released after an additional 100 min of incubation. Starch that remained unhydrolyzed after 120 min of incubation was considered resistant starch after fitting the results to an exponential equation, as explained by the referenced authors [7]. At the same time, total hydrolysis (TH) was measured.

#### 2.6.2. Digestion of Antioxidant Compounds

Simultaneously, during digestion, two aliquots (1 mL) were taken from the intestinal phase to determine TPC, TAC, and antioxidant capacity. The aliquot considered as a non-dialyzable fraction or that continues its passage to the colon was taken outside the dialysis membrane, and the aliquot considered as a dialyzable fraction or that is absorbed in the intestine, i.e., potentially bioavailable (PB) was taken from inside the dialysis membrane. The sum of these two fractions was considered as the bioaccessible (B) fraction.

### 2.7. Statistical Analysis

Data were examined using the InfoStat/Professional version 2020 software (Facultad de Ciencias Agropecuarias, Universidad Nacional de Córdoba). An Analysis of Variance (ANOVA) was performed, and the LSD Fisher comparison test was used, with a significance level of *p* < 0.05. Pearson’s correlation test was used to establish relationships between parameters (*p* < 0.05).

## 3. Results and Discussion

### 3.1. Flour Chemical Composition

The moisture content of flour was estimated on dry weight and ranged around 10%. The flour composition is shown in Table 1. The WM and PM flours did not show significant differences in lipid, ash, and carbohydrate content. Only protein content showed significant differences with a higher value for PM. Similar results were obtained in previous studies of the Moragro cultivar and other maize varietal types, such as opaque-2 [22], blue, and white maize [10]. CF did not show detectable effects of protein, lipids, and ash. However, it exhibited high carbohydrate content. Liu et al. [23] found similar trends in cassava and rice.

### 3.2. Viscosity and Flour Pasting Properties

Table 2 shows the pasting parameters measured in flours and flour blends. Cassava flour had the significantly highest PV and BD and the lowest PT among the flour samples. This is due to its higher starch content, which is composed of spheroidal and less compact granules, as suggested by Monthe et al. [24].

Whole purple maize flour exhibited significantly lower PV, BD, and SB than white maize, in agreement with Trehan et al. [25], who proposed that purple maize granules were more compact in the matrix, which allows for less diffusion of water inside the grain and lower amylose content. The same trend was observed in the decrease of SB, PV, and BD as the proportion of purple maize increased in flour blends. This decrease may be due to the replacement of cassava and rice with purple maize. Similar results were reported by Zheng et al. [26] when rice was replaced with Chinese berry leaves.

High anthocyanin concentration in flours may interfere with other molecules and produce undeveloped or underdeveloped doughs [27]. In that sense, a negative correlation was found in pre-fermented dough between total anthocyanin content with TV, SB and FV (r = −0.83 *p* < 0.01; r = −0.81 *p* < 0.05; r = −0.81 *p* < 0.05, respectively); between total polyphenol content with PV, TV, SB, and FV (r = −0.81 *p* < 0.05; r = −0.91 *p* < 0.01; r = −0.89 *p* < 0.01; r = −0.85 *p* < 0.01, respectively); and between ferulic acid with PV, TV, SB, FV (r = −0.88 *p* < 0.01; r = −0.82 *p* < 0.01; r = −0.8 *p* < 0.05; r = −0.81 *p* < 0.01, respectively). Similarly, a negative correlation was found in post-fermented dough between total anthocyanin content with TV, SB and FV (r = −0.82 *p* < 0.01; r = −0.73 *p* < 0.05; r = −0.79 *p* < 0.05, respectively); between total polyphenol content with PV, TV, SB, and FV (r = −0.74 *p* < 0.05; r = −0.88 *p* < 0.01; r = −0.79 *p* < 0.05; r = −0.86 *p* < 0.01, respectively); and between ferulic acid with PV, TV, SB and FV (r = −0.87 *p* < 0.01; r = −0.83 *p* < 0.01; r = −0.82 *p* < 0.01; r = −0.83 *p* < 0.01, respectively). In line with this, Zheng et al. [26] demonstrated that proanthocyanidins disrupt starch gelatinization and dough development in rice.

### 3.3. Effect of Bread Making on Anthocyanins, Polyphenols, Ferulic Acid and Antioxidant Capacity

#### 3.3.1. Raw Flours

Table 3 summarizes the content of bioactive compounds and antioxidant capacity determined in flours. Purple maize showed the highest values of TAC, TPC, FRAP, and TEAC, while FA was higher in White maize. Consistently, as the proportion of PM raised in formulation, the measured attributes also increased compared to the control. FA followed the same trend as the formulations with the addition of purple maize flour. However, the control showed higher FA than F34% PM. Other works reported higher values of TAC and TPC in purple maize genotypes [28,29]. WM, RF, and CF did not yield detectable levels of TAC and TEAC (Table 3). This result agrees with the study of Lopez-Martinez et al. [16] in white maize flour, coincides with the findings of Colombo et al. [30] in rice flour and also with those of Reinaldo et al. [31] in cassava flour. Trehan et al. [25] found similar FA levels for white and purple maize.

In a previous study of our research group [32], the main anthocyanin compounds found in Moragro whole flour were cyanidin derivatives: cyanidin-3-glucoside, cyanidin-3-(6” malonylglucoside) and cyanidin -3-(3”,6”, dimalonylglucoside). The most abundant non-anthocyanin compounds were caffeic acid, kaempferol 3-O-glucuronide, and citric acid. Similar results were found in other works [33].

#### 3.3.2. Effect of Kneading and Fermentation

During kneading, TAC was slightly modified in relation to raw flour, while a marked decrease in TPC and FA was observed (Figure 1). These results contradict Ktenioudaki et al. [34], who found that water, air, and energy supplied during the kneading process may break/degrade non-anthocyanin polyphenols, while anthocyanin compounds are more resistant to kneading. Our work proves that water and oxygen could activate the oxidative enzymes present in the flour and may affect phenolic compounds. In line with this, it has been reported that polyphenols could form complexes with starch or proteins and decrease the extractability during kneading [35].

The effect of fermentation was observed in the variations of pre-fermented to post-fermented doughs. In all formulations, a decrease in TAC was obtained, while a slight increase in FA was observed, and TPC slightly increased in doughs with 34% and 70% PM (Figure 1). The dough fermentation increased the release of polyphenols from fiber, which may result in higher extractability and antioxidant activity. Building on this, Angelino et al. [36] found that lower pH during fermentation promotes the activity of hydrolases and contributes to extensive hydrolysis of both esters and glycosides of phenolic acids. In addition, lower pH during fermentation promotes the activity of hydrolases and contributes to extensive hydrolysis of both esters and glycosides of phenolic acids. However, the lower TAC of the dough after fermentation could be due to changes in the pH or temperature of the environment, which would activate oxidative enzymes, therefore promoting the degradation of anthocyanins [37].

#### 3.3.3. Effect of Baking

The effect of cooking was observed through the variations of the post-fermented dough (prior to baking) to bread (Figure 1). After baking, TAC significantly decreased in all formulations, while TPC decreased in 34% and 50% PM bread and remained similar in 70% PM bread. Cooking did not markedly modify FA content, except for 70% PM bread, where the increase was significant. Several studies have found conflicting results about the effect of baking and temperature on bioactive compounds. However, many authors agree that heat stress is an important factor that influences the stability and bioavailability of bioactive compounds [38]. High temperatures could cause the degradation of conjugated polyphenolic compounds, which would result in an increase in free phenolic acids and an improvement of their bioavailability since free phenolics are more available than combined forms [34].

The process of baking in foods like muffins and cookies decreases the anthocyanins, mainly pelargonidin-3-O-glucoside and cyanidin-3-glucoside, because they are transformed into chalcones, resulting in low-stability compounds [37,38]. In addition, anthocyanins are susceptible to destabilization (degradation/discoloration) when exposed to factors like acid/alkaline, heat, light, oxygen, ascorbic acid, sulfite, metal ions, and enzymes. The anthocyanin stability is directly related to the level of glycosylation and indirectly influenced by the number of hydroxyl groups [39].

Conversely, the less marked variation in TPC compared to TAC after baking could also be due to anthocyanins and other compounds decomposing into products that can react with reactive Folin’s acid for its reducing capacity, as suggested by McDougall et al. [40].

Figure 2 shows the antioxidant capacity behavior of the bread-making process. The changes in the antioxidant activity of bread could be attributed to the changes in TPC and TAC, depending on the PM proportion in bread formulation (Figure 1). Similar results were reported by Hryhorenko et al. [41], who added different concentrations of wholegrain red sorghum flour in breads.

Baking significantly decreased the antioxidant activity in all formulations (Figure 2). Gu et al. [42] suggested that loss of antioxidant activity is a consequence of the phenolic structure alteration due to pressure and thermal treatment. However, Tian et al. [43] found that the baking process increased antioxidant activity in whole wheat breads. In our case, higher antioxidant activity was maintained in the bread with purple maize flour with respect to the control (Figure 2).

### 3.4. Bread Quality

#### 3.4.1. Specific Volume and Crumb Texture

Baked breads are shown in Figure 3. Bread formulations yielded significant differences in the bread quality parameters analyzed after baking; as the PM proportion increased, SV decreased, and firmness was higher (50% PM = 24.16 N ± 1.54 and 70% PM 46.26 N ± 0.25). In contrast, the 34% PM sample did not show a significant difference with the freshly baked control bread (7.3 N ± 0.55 and 8.32 N ± 0.11, respectively). Firmness measured at 24 and 72 h was lower in 34% PM than control bread (24 h: 16.99 N ± 0.57 vs. 23.24 ± 1.8; 72 h: 32.70 ± 0.73 vs. 43.45 ± 1.43, respectively), while 70% PM bread had firmer crumbs (90.24 N ± 5.84 and 100.38 N ± 7.73, respectively). Bread quality is associated with the dough’s rheological properties and its ability to trap and hold the gas in the enclosed cells and expand [44]. Gas retention is primarily controlled by the viscosity of the batter [11]. Higher starch content promotes higher gelatinization, which contributes to the formation of a cohesive network capable of retaining gas [45]. However, lower gelatinization decreases the dough’s mechanical strength and results in lower consistency [46]. In line with our study, a positive relationship was found between bread SV and the viscosity parameters measured in the flour blends, such as PV (r = 0.98 *p* < 0.001), BD (r = 0.89 *p* < 0.01), FV (r = 0.98 *p* < 0.001), SB (r = 0.96 *p* < 0.001). The volume and expansion achieved after fermentation and baking can also be affected by the way in which the different starchy substances are compacted [47]. However, flour polyphenols could compete with water by compromising starch swelling, delaying gelatinization, and reducing the viscosity [26]. In our study, as the PM proportion increased, the dough viscosity decreased (Table 2), with a consequent decrease in bread-specific volume (Figure 3). The presence of anthocyanins and their numerous hydroxyl groups can form hydrogen bonds with amylose, inhibiting the amylose leaching and reducing the viscosity [48].

#### 3.4.2. Starch Digestibility of Breads

Starch digestibility parameters and digestibility curves are shown in Table 4 and Figure 4, respectively. The lowest RDS value was 70% PM, followed by 34% PM, while the control showed the highest value. An opposite behavior was observed in RS, although bread of 34% and 50% PM did not significantly differ (Figure 4). High RS values in the 34% and 50% PM bread could contribute to gastrointestinal health and favor microbial fermentation of the large intestine, as suggested by Macfarlane and Macfarlane [49]. Bread of 34% PM presented the highest SDS, followed by 50 and 70% PM (Table 4). Other works [9,10] reported that the higher RS and SDS and the lower RDS values in cooked pigmented maize -with respect to white maize- may be attributed to the molecular structure of purple maize starch and to anthocyanin-starch interactions, which can inhibit or delay enzymatic activity. Aalim et al. [50] have shown that enzyme inhibition, like α-amylase, by the presence of phenolic compounds, causes the reduced and retarded digestibility of starch chains. In this sense, our study revealed a positive correlation between the percentage of PM replacement and RS content (r = 0.81 *p* < 0.05). The interaction between polyphenols and starch increased the RS fraction due to the alteration of the microstructural starch arrangements, affecting digestibility and retarding the gelatinization [51].

Purple maize breads showed significantly lower Total Hydrolysis than control bread, although those with 34% and 50% PM did not show differences between them (Table 4). In addition, the behavior of TH was similar to RDS, which is consistent with other studies [52].

#### 3.4.3. In Vitro Digestibility of Antioxidant Compounds of Breads

Table 5 shows the values of bioactive compounds and antioxidant capacity after bread digestion. TAC was not detected, and neither were bioaccessible and potentially bioavailable fractions in control bread. Bread with 70% PM showed significantly higher TAC than the 34 and 50% PM bread, which did not show differences between them (Table 5). However, when the purple maize breads were digested, TAC was not detected in B and PB fractions, indicating that anthocyanins present in breads were degraded during digestion. Similar results have been reported by Herrera-Balandrano et al. [53], who suggested that the most prominent anthocyanin loss occurs in the intestinal phase.

Significant differences were obtained in the TPC of bread, with higher contents depending on higher PM proportions. However, after digestion, purple maize bread suffered a loss of TPC, and the PB fraction was significantly lower than the B fraction. Only the control formulation showed a slightly higher TPC of B fraction than bread (Table 5), probably due to the influence of the food matrix and the polyphenol chemical structure of white maize, as suggested by Sęczyk et al. [54]. In purple maize bread, the TPC of PB fractions was between 13 and 23% of B fractions for all formulations. Several studies reported a decrease in total polyphenol content after food digestion [55]. The polyphenol bioavailability may be attributed to their specific chemical structures, such as molecular weight, glycosylation, and esterification [56], as well as structural changes generated by digestion conditions [57]. However, polyphenols from the bioaccessible fraction that are not bioavailable can reach the colon and be used by the microbiota, exerting their beneficial action there [58].

FRAP of B fraction values followed a similar trend to TPC, except for the formulation with 34% PM and control, where B fractions exceeded those found in bread. However, the PB fraction was lower than B in all formulations, both in purple maize and control bread (Table 5). The decrease in the reducing capacity of the two fractions may be due to TPC changes in the gastrointestinal tract [59]. Several studies demonstrated similar results, indicating that lower antioxidant capacity is related to the decrease in phenolic compounds [55,60].

The TEAC detected in the B fraction was higher than in the PB fraction and bread in all PM formulations (Table 5), probably because the digestion generates a transition from an acidic to an alkaline medium that favors the deprotonation of aromatic ring hydroxyl groups of phenolic compounds [61]. This reaction could also be due to the fact that bound phenolic compounds or some other bioactive components might be more efficient at scavenging ABTS radicals [62]. The TEAC of the PB fraction was significantly lower in PM formulations, which is consistent with TPC and FRAP (Table 5). Polyphenols may not be able to pass through the dialysis tubing due to polarity, size, or interaction with the food matrix [63] and, therefore, cannot exert their antioxidant activity.

It should be noted that post-digestion TPC and TEAC increased as purple maize flour was supplemented. The slight discrepancy found between FRAP and TEAC values can be attributed to different antioxidant capacity mechanisms evaluated by both assays [64].

## 4. Conclusions

The addition of whole purple maize flour increased the content of polyphenols and anthocyanins in gluten-free bread, although the fermentation and baking processes generally resulted in a loss of these compounds. Bread made with 34% of purple maize showed a higher SV within the purple maize bread, and it presented an increase in antioxidant capacity with respect to the control bread. The TPC and TAC of purple maize interfered with starch gelatinization and, consequently, decreased dough consistency and bread-specific volume. However, these compounds influenced bread starch digestibility, decreasing RDS and increasing RS, which is nutritionally favorable.

The digestive processes and the dialyzability study, in general, decreased polyphenol compounds. Nevertheless, the antioxidant capacity of the bioaccessible fraction was higher even than bread, although a lower number of polyphenols and antioxidant capacity remained potentially bioavailable. This indicates that whole purple maize flour represents a potential ingredient for gluten-free bread production with an improved nutritional profile and functional properties. Specifically, bread with 34% purple maize in formulation shows acceptable volume, lower starch digestibility, and contribution of bioactive compounds with antioxidant capacity.

## Figures and Tables

**Figure 1 foods-13-00194-f001:**
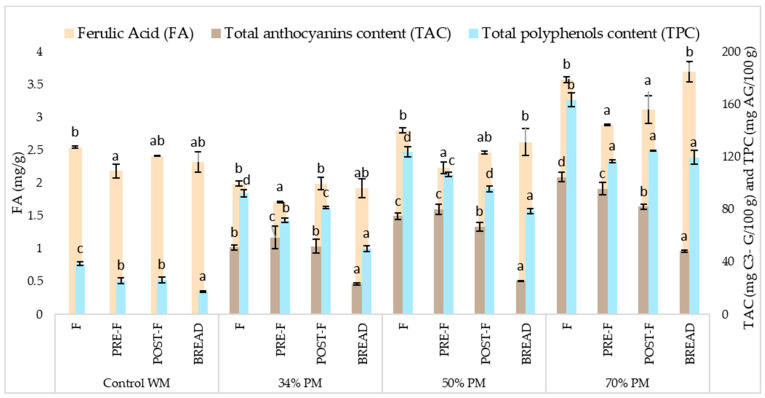
Total anthocyanin, polyphenol, and ferulic acid content in different stages of purple maize gluten-free bread making. Values followed by different letters are significantly different for the same formulation (*p* < 0.05). Control WM: Formulation of control with white maize; 34% PM: Formulation with 34% purple maize; 50% PM: Formulation with 50% purple maize; 70% PM: Formulation with 70% purple maize. F: Flour; PRE-F: Pre-fermented dough; POST-F: Post-fermented dough. FA: Ferulic acid; TAC: Total anthocyanin content; TPC: Total polyphenol content.

**Figure 2 foods-13-00194-f002:**
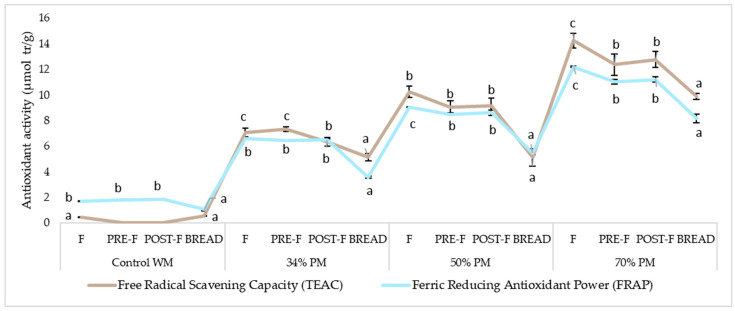
Antioxidant activity in the different stages of purple maize gluten-free bread making. Values followed by different letters are significantly different for the same formulation (*p* < 0.05). Control WM: Formulation control with white maize; 34% PM: Formulation with 34% purple maize; 50% PM: Formulation with 50% purple maize; 70% PM: Formulation with 70% purple maize. F: Flour; PRE-F: Pre-fermented; POST-F: Post-fermented.

**Figure 3 foods-13-00194-f003:**
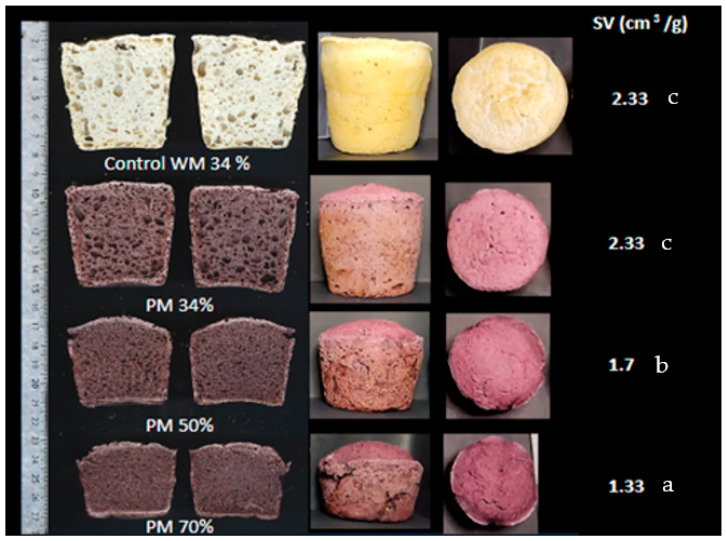
Specific volume of breads made with purple maize flour. Values followed by different letters are significantly different (*p* < 0.05). SV: Specific volume. Control WM: Formulation Control with white maize; 34% PM: Formulation with 34% purple maize; 50% PM: Formulation with 50% purple maize; 70% PM: Formulation with 70% purple maize.

**Figure 4 foods-13-00194-f004:**
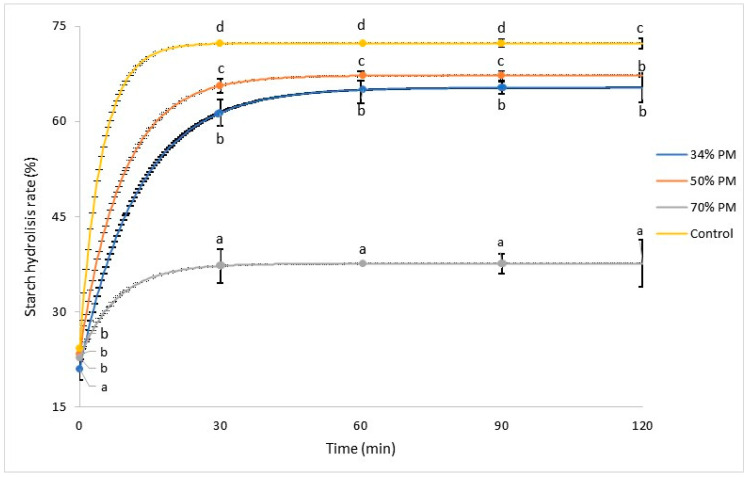
Starch digestibility curves of gluten-free breads with different proportions of purple maize. Values followed by different letters are significantly different (*p* < 0.05). 34% PM: formulation with 34% purple maize; 50% PM: formulation with 50% purple maize; 70% PM: formulation with 70% purple maize.

**Table 1 foods-13-00194-t001:** Chemical components of the raw material.

Flour ^1^	Protein (%)	Lipids (%)	Ash (%)	Carbohydrates (%)
PM	10.08 ± 0.56 ^b^	3.78 ± 0.11 ^b^	1.78 ± 0.09 ^b^	84.45 ± 0.60 ^a^
WM	8.44 ± 0.12 ^a^	4.42 ± 0.29 ^b^	1.69 ± 0.02 ^b^	85.45 ± 0.17 ^a^
CF	nd	nd	nd	99.51 ± 0.18 ^c^
RF	9.24 ± 0.25 ^ab^	2.25 ± 0.17 ^a^	1.51 ± 0.04 ^a^	86.99 ± 0.26 ^b^

Values followed by different letters in the same column are significantly different (*p* < 0.05). ^1^ PM: Purple maize flour; WM: White maize flour; CF: Cassava flour; RF: Rice flour; nd: not detected.

**Table 2 foods-13-00194-t002:** Viscosity and pasting properties parameters of raw flours and flour blends ^1^.

Flour ^2^	PV (cP)	TV (cP)	BD (cP)	FV (cP)	SB (cP)	PT (°C)
Rice	3759.0 ± 44.3 ^f^	2471.3 ± 32.6 ^i^	1287.7 ± 36.7 ^e^	5193.7 ± 39.8 ^j^	2722.3 ± 54.1 ^g^	81.2 ± 0.4 ^g^
Cassava	6380.0 ± 88.6 ^h^	2101.0 ± 17.4 ^h^	4279.0 ± 83.5 ^g^	2970.2 ± 42.0 ^h^	869.2 ± 51.2 ^c^	71.2 ± 0.3 ^a^
WM	2353.0 ± 102.4 ^e^	1147.7 ± 22.7 ^g^	1205.3 ± 122.0 ^e^	2797.3 ± 57.0 ^g^	1649.7 ± 60.6 ^e^	77.8 ± 0.5 ^f^
PM	1049.0 ± 27.0 ^b^	536.3 ± 18.8 ^a^	512.6 ± 9.5 ^d^	1255.3 ± 44.8 ^a^	719.0 ± 26.4 ^a^	76.9 ± 0.4 ^e^
34% WM	1306.3 ± 37.2 ^c^	889.0 ± 18.0 ^ef^	417.3 ± 22.8 ^c^	1769.0 ± 31.7 ^e^	880.0 ± 14.7 ^c^	74.3 ± 0.1 ^c^
34% PM	1409.0 ± 18.0 ^d^	866.6 ± 7.3 ^de^	542.3 ± 17.3 ^d^	1754.7 ± 14.4 ^e^	888.0 ± 11.5 ^cd^	74.5 ± 0.1 ^c^
50% CF-50% RF	4158.7 ± 26.4 ^g^	2626.7 ± 50.2 ^j^	1532.0 ± 27.7 ^f^	4455.3 ± 47.0 ^i^	1828.7 ± 63.5 ^f^	74.0 ± 0.4 ^bc^
50% WM	1095.3 ± 9.8 ^b^	830.3 ± 2.3 ^d^	265.0 ± 8.8 ^a^	1654.3 ± 7.0 ^d^	824.0 ± 5.2 ^bc^	74.2 ± 0.1 ^c^
50% PM	1021.3 ± 20.9 ^b^	709.3 ± 10.5 ^c^	312.0 ± 11.2 ^ab^	1489.0 ± 22.0 ^c^	779.6 ± 11.5 ^ab^	73.4 ± 0.1 ^b^
70% WM	1318.0 ± 34.7 ^c^	923.6 ± 30.6 ^f^	394.3 ± 4.1 ^bc^	1876.0 ± 56.5 ^f^	952.3 ± 26.1 ^d^	75.4 ± 0.4 ^d^
70% PM	876.0 ± 38.6 ^a^	603.3 ± 29.9 ^b^	272.6 ± 10.2 ^a^	1355.7 ± 69.3 ^b^	752.3 ± 39.5 ^a^	74.5 ± 0.4 ^c^

Values followed by different letters in the same column are significantly different (*p* < 0.05). ^1^ PV: Peak viscosity; TV: Trough viscosity; BD: Breakdown; FV: Final viscosity; SB: Setback; PT: Pasting temperature. ^2^ WM: Raw white maize flour; PM: Raw purple maize flour; 34% WM: Flour blend with 34% white maize, 33% rice, and 33% cassava; 34% PM: Flour blend with 34% purple maize 33% rice, and 33% cassava; 50% CF-50% RF: Flour blend with 50% cassava and 50% rice flour; 50% WF: Flour blend with 50% white maize, 25% rice and 25% cassava; 50% PM: Flour blend with 50% purple maize, 25% rice, and 25% cassava; 70% WM: Flour blend with 70% white maize, 15% rice, and 15% cassava; 70% PM: Flour blend with 70% purple maize, 15% rice, and 15% cassava.

**Table 3 foods-13-00194-t003:** Polyphenolic compounds and antioxidant activity ^1^ of flours and formulations.

Flour ^2^	TAC (mg c3-GE/100 g)	TPC (mg AG/100 g)	FRAP (µmol tr/g)	TEAC (µmol tr/g)	FA (mg/g)
RF	nd	49.24 ± 1.15 ^b^	2.39 ± 0.03 ^b^	nd	1.39 ± 0.05 ^a^
CF	nd	nd	nd	nd	nd
WM	nd	65.73 ± 5.34 ^c^	2.59 ± 0.04 ^c^	nd	5.80 ± 0.03 ^h^
PM	149.27 ± 5.12 ^d^	222.39 ± 7.39 ^g^	16.88 ± 0.07 ^g^	20.21 ± 0.76 ^d^	4.73 ± 0.04 ^g^
CONTROL ^3^	nd	38.60 ± 1.58 ^a^	1.71 ± 0.01 ^a^	nd	2.54 ± 0.01 ^c^
F34% PM	50.87 ± 1.78 ^a^	91.86 ± 2.80 ^d^	6.57 ± 0.01 ^d^	7.06 ± 0.34 ^a^	1.99 ± 0.03 ^b^
F50% PM	74.72 ± 2.59 ^b^	123.50 ± 3.91 ^e^	9.07 ± 0.02 ^e^	10.25 ± 0.44 ^b^	2.80 ± 0.04 ^d^
F70% PM	104.54 ± 3.60 ^c^	163.00 ± 5.30 ^f^	12.20 ± 0.04 ^f^	14.24 ± 0.57 ^c^	3.57 ± 0.04 ^e^

Values followed by different letters are significantly different (*p* < 0.05). ^1^ TAC: Total anthocyanins content; TPC: Total polyphenols content; FRAP: Ferric reducing antioxidant power; TEAC: Trolox equivalent antioxidant capacity; FA: Ferulic acid; nd: not detected. ^2^ RF: Raw rice flour; CF: Raw cassava flour; WM: Raw white maize flour; PM: Raw purple maize flour. ^3^ Each formulation (CONTROL, F34% PM, F50% PM, and F70% PM) was calculated proportionally to the share of each raw flour.

**Table 4 foods-13-00194-t004:** Starch digestibility parameters ^1^ of gluten-free bread with different proportions of purple maize flour.

Formulation ^2^	RDS (g/100 g)	SDS (g/100 g)	RS (g/100 g)	TH (%)
34% PM	56.39 ± 0.05 ^b^	8.90 ± 1.76 ^c^	34.70 ± 1.81 ^b^	65.30 ± 1.81 ^b^
50% PM	62.33 ± 0.72 ^c^	4.90 ± 0.62 ^b^	32.77 ± 0.09 ^b^	67.23 ± 0.09 ^b^
70% PM	36.52 ± 0.01 ^a^	2.47 ± 1.66 ^ab^	61.01 ± 1.68 ^c^	38.99 ± 1.68 ^a^
Control	71.60 ± 0.52 ^d^	0.72 ± 0.02 ^a^	27.68 ± 0.55 ^a^	72.32 ± 0.55 ^c^

Values followed by different letters in the same column are significantly different (*p* < 0.05). ^1^ RDS: Rapidly digestible starch; SDS: Slowly digestible starch; RS: resistant starch; TH: Total hydrolysis. ^2^ 34% PM: formulation with 34% purple maize; 50% PM: formulation with 50% purple maize; 70% PM: formulation with 70% purple maize.

**Table 5 foods-13-00194-t005:** Bioactive compounds and antioxidant capacity after digestion of purple gluten-free maize bread ^1^.

		Control	34% PM ^2^	50% PM	70% PM
TAC (mg C3-G/100 g)	BREAD ^3^	nd	23.14 ± 0.97 ^a^	25.54 ± 2.63 ^a^	48.08 ± 2.05 ^b^
	B	nd	nd	nd	nd
	PB	nd	nd	nd	nd
TPC (mg AG/100 g)	BREAD	17.39 ± 0.66 ^Ba^	49.96 ± 2.25 ^Cb^	78.68 ± 1.85 ^Cc^	119.29 ± 5.30 ^Cd^
	B	22.16 ± 0.73 ^Ca^	24.93 ± 0.84 ^Bab^	27.31 ± 0.84 ^Bb^	50.16 ± 2.80 ^Bc^
	PB	4.72 ± 0.13 ^Aa^	5.33 ± 0.20 ^Aab^	6.39 ± 0.52 ^Abc^	6.78 ± 0.56 ^Ac^
FRAP (μmol TR/g)	BREAD	1.05 ± 0.10 ^Ba^	3.56 ± 0.07 ^Bb^	5.45 ± 0.13 ^Cc^	8.17 ± 0.33 ^Cd^
	B	3.27 ± 0.43 ^Ca^	3.88 ± 0.21 ^Ca^	3.85 ± 0.12 ^Ba^	7.55 ± 0.09 ^Bb^
	PB	0.28 ± 0.04 ^Aa^	0.37 ± 0.03 ^Aa^	0.49 ± 0.20 ^Aa^	0.53 ± 0.10 ^Aa^
TEAC (μmol TR/g)	BREAD	0.55 ± 0.02 ^Aa^	5.12 ± 0.29 ^Bb^	5.53 ± 0.60 ^Bb^	9.87 ± 0.23 ^Bc^
	B	10.20 ± 0.10 ^Ca^	10.85 ± 0.11 ^Cb^	11.00 ± 0.01 ^Cbc^	11.13 ± 0.04 ^Cc^
	PB	3.46 ± 0.01 ^Ba^	3.68 ± 0.03 ^Ab^	3.70 ± 0.01 ^Ab^	3.76 ± 0.01 ^Ac^

Values followed by different capital letters indicate significant differences (*p* < 0.05) within the same column. Values followed by lowercase letters indicate significant differences (*p* < 0.05) within the same row. ^1^ TAC: Total anthocyanins content; TPC: Total polyphenols content; FRAP: Ferric reducing antioxidant power; TEAC: Trolox equivalent antioxidant capacity. nd: not detected. ^2^ 34% PM: Formulation with 34% purple maize; 50% PM: Formulation with 50% purple maize; 70% PM: Formulation with 70% purple maize. ^3^ BREAD: measurements performed in bread; B: bioaccessible (sum of the dialyzable and non-dialyzable fraction); PB: potentially bioavailable (dialyzable fraction).

## Data Availability

The data presented in this study are available upon request from the corresponding author. The data are not publicly available due to being protected by intellectual property.

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
