# Peer review of "Whole Flour of Purple Maize as a Functional Ingredient of Gluten-Free Bread: Effect of In Vitro Digestion on Starch and Bioaccessibility of Bioactive Compounds"

_foods, 2024, doi:10.3390/foods13020194_

Round 1

Reviewer 1 Report

Comments and Suggestions for Authors

This study describes the effect of purple maize flour supplementation on the viscosity and pasting properties, starch digestibility, as well as phenolics contents and antioxidant activity affected by in vitro digestion. The search for novel functional ingredients rich in bioactive compounds for food fortification follows the current trends in food technology. Although there are many studies in the area of bread fortification, the topic is quite interesting, especially the determination of starch digestibility, phytochemical contents and antioxidant activity under artificial digestion conditions. Nevertheless, in my opinion, included in the title term „bioavailability”, which indicates part of the substance released during digestion and absorber (!), should be reserved for in vivo or cell line studies, therefore, in this case, the more suitable will be therm „bioaccessibility”. The abstract well reflects the content of the manuscript. The introduction provides a good background of the study and includes relevant references. The experiments, despite some having minor shortcomings, in overall well described. The experiment design could have been more carefully planned (i.e. HPLC determination of ferulic acid instead of spectrophotometrical). Furthermore, the modes of results presentation should be improved (details are described below). The discussion is supported by the obtained results and the conclusion summarizes the most important findings. Due to the described shortcomings influencing the quality of the manuscript I recommended the major revision.

Detailed suggestions:

Title - in accordance with the statements in lines 47-50, the term “Bioaccessibility” (or even „in vitro bioaccessibility”) more properly describes measured features and should be used in the title and throughout the manuscript. Performed dialysis allows to obtaining low molecular fraction which is considered as potentially available for absorption (not absorbed!). Due to the polyphenols can be absorbed in native forms (via passive diffusion, which is commonly regarded as the mechanism of polyphenol absorption), dialysis allows separate them from complexes e.g. with enzymes proteins and other food matrix components.  

lines 47-50  Bioaccessibility is defined as the maximum fraction of a substance that can be dissolved and released in the gastrointestinal tract from the original food matrix [6]. Bioavailability is understood as the fraction of a substance that is absorbed by the gastrointestinal tract and reaches the systemic circulation [7].

Line 146-155 – In future studies consider using HPLC for the determination of rosmarinic acid instead of spectrophotometric measurement, as the more precise method. It could provide more specific results.

Line 143-174 – “Finally, a dialysis membrane (MWcutoff=10 kDa) with 1M of sodium
bicarbonate was added.” Underline the use of 1M of sodium
bicarbonate for dialysis. It is commonly known that an alkaline environment (provided by 1M of sodium bicarbonate) can induce polyphenol substances degradation and observed effects after dialysis - a decrease of polyphenol contents could result in polyphenols degradation instead of their permeability throughout the membrane. The stability was checked in such conditions? Why authors didn’t use distilled water?

Line 162 – „2.6. In vitro static digestion of starch and antioxidant compounds” applied during digestion enzymes does not allow the digestion of antioxidant compounds (it is possible in some cases by using e.g.micriobial esterases) therefore, the statement digestion of antioxidant compounds seems inappropriate. Maybe  „2.6. In vitro static digestion”?

Lines 177 – 183 – I recommended separating this fragment into separate subsections e.g. 2.6.1 analysis of starch fractions/ digestibility or similar. The description of these methods in one section is confusing.

Line 175 – „The reducing sugar content was determined by the 3,5-dinitrosalicylic acid (DNS) method.” Add reference or describe this method. Similarly line 183.

Lines 177-178 – Why starch digestibility determination was not performed using obtained fractions after the INFOGEST procedure?

Line 206 – “effects” instead of „values” seems more suitable

Line 222, 258, 382, 435 – Tables – Using superscript letters for indication of statistically significant differences increases the readability of results. Add standard deviations for results presented means.  Unify decimal places for the same analysis. Table 2 – unify abbreviation nd (nd or Nd). Tables 3 and 4 – use dots instead of commas for decimals

Line 279, 327, 421 – Figures. Add lower error bars.

Figure 4 – add lower and upper error bars, indicating statistical significance.

Figures 3 and 4 – figures should be placed near the text describing them. Change the placement of Figures 3 and 4.  For example, it is confusing, when describing the antioxidant properties in section 3.4.3 (line 392) are placed results (figure) from section 3.4.2. (line 359). It should be ordered.

Reviewer 2 Report

Comments and Suggestions for Authors

This paper deals with whole flour of purple maize as functional ingredient of gluten free breads, the experiments were well designed, and the materials used for bread were well analyzed. The conclusion is that, the breads with 34% PM show acceptable volume, lower starch digestibility and contribution of bioactive compounds with antioxidant capacity. The feasibility of bread with purple maize flours should be strengthened.

Suggestion,

1)     The decimal point in the whole paper should not be a comma;

2)     Values followed by different letters are significantly different (P<0.05), the letters should be small letters. In statistics, if capital letters are used, the significant levels are at P<0.01. In all the tables, the data should be mean±standard errors.

3)     There are many abbreviations. An abbreviation table should be given.

4)     In the abstract, FRAP and ABTS should be noted. Line 14 to 28, only four lines described the results and conclusion, the results had no description at all, and new findings did not be given. The introduction of experimental methods occupies more content.

5)     Nutrition and health are the themes of today’s world, but the feasibility of bread with purple maize flours should be given. Although the authors showed the specific volume of breads, the textural parameters of breads should be given in this paper.

6)     Fig. 4 should have standard errors.

7)     Line 173, the dialysis time should be given.

8)     The pancreatic alpha-amylase and amyloglucosidase should be noted the company resource.

9)     Line 140 to 143, the calculation formula of total anthocyanin content should be given.

10)  Line 198 to 209, it had better give a table to show the chemical composition of food materials.

Round 2

Reviewer 1 Report

Comments and Suggestions for Authors

The necessary explanations were provided and the manuscript has been sufficiently improved to meet the basic requirements of the journal.